# Pulse Radar with Field-Programmable Gate Array Range Compression for Real Time Displacement and Vibration Monitoring

**DOI:** 10.3390/s19010082

**Published:** 2018-12-27

**Authors:** Mihai-Liviu Tudose, Andrei Anghel, Remus Cacoveanu, Mihai Datcu

**Affiliations:** 1Research Centre for Spatial Information–CEOSpaceTech, University “Politehnica” of Bucharest, Bucharest 011061, Romania; andrei.anghel@munde.pub.ro (A.A.); remus.cacoveanu@upb.ro (R.C.); mihai.datcu@dlr.de (M.D.); 2German Aerospace Centre, Remote Sensing Technology Institute, Weßling 82234, Germany

**Keywords:** pulse radar, real time, displacement, vibration, FPGA, USRP, cross-correlation

## Abstract

This paper aims to present the basic functionality of a radar platform for real time monitoring of displacement and vibration. The real time capabilities make the radar platform useful when live monitoring of targets is required. The system is based on the RF analog front-end of a USRP, and the range compression (time-domain cross-correlation) is implemented on the FPGA included in the USRP. Further processing is performed on the host computer to plot real time range profiles, displacements, vibration frequencies spectra and spectrograms (waterfall plots) for long term monitoring. The system is currently in experimental form and the present paper aims to prove its functionality. The precision of this system is estimated (using the 3σ approximation) at 0.6 mm for displacement measurements and 1.8 mm for vibration amplitude measurements.

## 1. Introduction

While there is a great variety of methods for vibration monitoring of remote targets, most of them require the sensor to be mounted on the desired part, which may prove inconvenient in several applications. The non-contact vibration monitoring methods are usually based on radar and laser, but the latter suffers from atmospheric influence and requires a certain degree of target reflectivity in the optical domain.

Ground-based radars are being used frequently in the present days to measure target displacements, especially for structural monitoring. By means of Doppler effect or by means of interferometry, radars and also sonars can also be used in non-contact vibration monitoring. In the field of sonar, target motion analysis has also been performed using probabilistic techniques, based on the amplitude information of the target [1]. The method increases the accuracy of the estimation and the observability of the target. It is meant for narrowband passive sonar tracking system. A study about the usage of the target-related information with the purpose of determining the trajectory of the observing platform was published [2]. It is useful if the information transmitted by the platform is intercepted.

Vibration monitoring is useful especially for medical purposes (patient breath and heartbeat monitoring, detection of life-signs) [3,4], civil engineering, structural health monitoring (bridges and dams monitoring) [5,6] and industrial engineering (monitoring of machine parts vibrations) [7,8]. A theoretical study was written [9] about the pulse radar counterpart, the FMCW radar, in order to evaluate its performances in the context of vibration measurement. The authors state that it can achieve sub-Hertz theoretical accuracy of the vibration frequency measurement and micrometer accuracy for mechanical oscillation amplitude measurement. In [10], a study about the effects of the antenna radiation pattern in the context of vibration monitoring using radar was published.

Most of the mentioned systems offer the information about displacements and vibration after the acquisition of data and post-processing. In contrast, the system presented in this paper contains the implementation of the FPGA baseband part for signal processing in a pulse radar system, with real time range compression. Range compression is the process of generating the range profile using the transmitted and received signals. 

Reference [11] describes the signal processing parts required for real time vibration monitoring of targets, using an FMCW radar system. The information about displacement or vibration is extracted from the interferometric phase of the beat signal. The authors provide test results regarding measurements at low vibration frequencies and at very short range (1.45 m). The results presented in this paper cover higher vibration frequencies on targets placed at longer ranges.

In reference [12], a SAR processor for single-bit coded signals is presented. It includes the time-domain range and azimuth compression parts. This SAR processor was designed for the C-band airborne pulse radar called “E-SAR”. It is implemented in FPGA, due to low power and low area consumption. The single bit architecture allows fundamental simplification, but still serves the purpose of range compression, similar to our own system.

A more recent data processor [13] was designed for precipitation radar on-board data processing. As in our case, the transmitted waveform is a chirped impulse. The platform performs the range compression by matched filtering, using a finite impulse response filter, implemented in FPGA. The mentioned filter has a number of 256 taps and uses most of the FPGA resources by performing a number of 20 billion operations per second. Since the filter has fixed coefficients, range compression is always performed between the received signal and a reference signal. In this configuration, the radar system is required to be phase coherent from one transmitted pulse to another. In contrast, our implementation uses a reference signal which is a sampled replica of the transmitted signal. Therefore, it requires phase coherence only for one pulse repetition period.

Another design of a UAV-borne SAR processor is described in [14], which is meant for natural hazard detection. The solution is based on combining a dedicated preprocessor with an FPGA. Tasks are divided between the two, where the preprocessor executes irregular computations and the FPGA performs repetitious computations. Range compression is also performed on the FPGA.

The previous three presented systems feature integrated FPGA pulse compression and are meant, respectively, for SAR airborne imaging, precipitation monitoring and natural hazard detection. The system described in this paper is meant for displacement and vibration monitoring.

Section 2 of this paper presents the principle of pulse radar displacement measurement using interferometry. An overview description of the displacement and vibration measurement system is found in Section 3. Section 4 of this paper describes the FPGA hardware design implementation of the radar’s transmit and receive baseband modules. Range compression through cross-correlation implementation on the FPGA is described in Section 5. The functions performed by the host computer are presented in Section 6. In-field experimental results are shown in Section 7, while the conclusions can be found in Section 8.

**Notations**: *t_del_*: radar-to-target round trip delay, *c*: speed of light, *R*_0_: radar-to-target fixed range component, Δ*R*(*t*): radar-to-target variable range component, *τ*: chirp duration, *α*: chirp angular rate, *f*_0_: carrier frequency, *φ*_0_: initial phase, *δφ*: phase shift, *K*: number of simultaneous multiplications, *N*: length of the input sequence, *δr*: range bin width, *σ_displ_*: displacement standard deviation, *λ*: wavelength, *SNR*: Signal-to-Noise Ratio.

## 2. Principle of Pulse Radar Displacement Monitoring

The geometry of the displacement measurement system is presented in Figure 1.

The round-trip delay term *t_del_* is dependent on the radar-to-target range, with expression (1):(1)$$t_{del}=\frac{2\left(R_{0}+\Delta R\left(t\right)\right)}{c}$$
where *R*_0_ is the fixed range component, and Δ*R*(*t*) is the variable range component, which describes displacement or vibration.

The radar transmits a linearly frequency modulated pulse of *τ* duration, with the time-domain expression:(2)$$s_{\mathrm{TX}}\left(t\right)=\mathrm{rect}\left(\frac{t}{\tau }-\frac{1}{2}\right)\exp\left[j\left(2\pi f_{0}t+\frac{\alpha t^{2}}{2}+\varphi_{0}\right)\right]$$
where rect(t) is the rectangular function (defined as 0 for |*t*| > 1/2, ½ for *t* = ½, and 1 for |*t*| < 1/2), *τ* is the chirp duration, *α* is the chirp angular rate, *f*_0_ is the carrier frequency and *φ*_0_ is the initial phase. 

The received signal is delayed by a time duration proportional to the signal time of flight *t_del_*.
(3)$$s_{\mathrm{RX}}\left(t\right)=\mathrm{rect}\left(\frac{t-t_{del}}{\tau }-\frac{1}{2}\right)\cdot \exp\left[j\left(2\pi f_{0}\left(t-t_{del}\right)+\frac{\alpha {\left(t-t_{del}\right)}^{2}}{2}+\varphi_{0}\right)\right]$$

The signals are transferred to baseband by frequency mixing with *f*_0_.

The range compressed signal is found by determining the cross-correlation between the baseband versions of the transmitted and received signal.
(4)$$r_{\mathrm{TX}–\mathrm{RX}}\left(t\right)=\tau \left(1-\frac{\left|t-t_{del}\right|}{\tau }\right)\exp\left[j\left(-2\pi f_{0}t_{del}\right)\right]\mathrm{psf}\left(t-t_{del}\right)$$
where psf() is the range point spread function (the autocorrelation function of the complex envelope) [15].

Expanding the previous relation yields:(5)$$r_{\mathrm{TX}–\mathrm{RX}}\left(t\right)=\tau \left(1-\frac{\left|t-t_{del}\right|}{\tau }\right)\mathrm{psf}\left(t-t_{del}\right)\cdot \exp\left(-j\frac{4\pi f_{0}\left(R_{0}+\Delta R\left(t\right)\right)}{c}\right)$$

The phase of the last term is important in displacement determination. If the phase shift *δφ* can be measured in the slow-time domain (from one pulse to another), then:(6)$$\Delta R\left(t\right)=\frac{c}{4\pi f_{0}}\delta \varphi$$

Therefore, the variable range component can be determined by measuring the phase in the bin corresponding to the target in the range profile.

## 3. System Overview

The radar system in Figure 2 is implemented using an “USRP-2954R” platform and a host computer. The “Universal Software Radio Peripheral” is a flexible platform, containing RF transmit and receive modules, as well as an FPGA used for digital baseband signal processing. The processing part from the host computer is performed with “LabView”. The digital baseband part is executed within an FPGA, which is programmed with code compiled using “LabView FPGA”.

The system contains the digital transmit and receive parts, as well as the block that computes the range compression using cross-correlation. Practically, the aim is the fast computation of the range profile as soon as the received signals samples have been stored in memory. The system is based on split processing of signals, which occurs both on the FPGA, as well as on the host computer. The processing required for range compression (cross correlation) is hardware implemented on the FPGA for increased speed. Based on the delivered range profiles, fast Fourier transform (FFT) for vibration spectrum estimation is performed by the host computer. This combined solution offers real time target displacement and vibration measurements.

The developed graphical interface allows the user to select the target bin of interest and monitor it separately for displacement and vibration analysis. While a single bin can be monitored independently, the magnitude and phase components of the entire range profile are always available, offering a glance of the target scene at any moment.

The transmitted signal path is connected to the input of a splitter, which divides it between the RX port of the first daughterboard and the transmit path. This signal is used as a reference, in order to determine the time instance at which the transmit starts, therefore cancelling the group delay time of the analog RF circuits.

Since the implementation requires two input signals in order to compute the range profile, this configuration can be used as monostatic radar, with the on-board transmitter, or as bistatic radar with a transmitter of opportunity [15]. In the bistatic radar configuration, the transmit part can be disabled and both the RX channels can be used: one for the reference signal (directly from the transmitter of opportunity) and the second one for the reflected signal from the scene, although careful triggering will be required in this case. 

## 4. Pulse Radar Baseband Implementation on the USRP Platform

This section contains a description of the baseband part of this pulse radar. The block diagram in Figure 3 shows the main components, consisting of counters, memories and a specific block for computing cross correlation, which will be described in more detail in the next section.

A counter that is incremented at each sampling clock period (120MHz) is used in order to generate the pulse repetition interval trigger (“Start trigger 1” in Figure 3). The bits composing the address number are tied to a read only memory and to two additional read/write memories. All of the memories contain both I and Q samples. The TX read only memory contains the samples of a chirp signal. 

Practically, as the counter increments the value of the address location, the samples from the TX memory are delivered to the digital to analog converter (DAC) and the samples from the two analog to digital converters (ADCs) are written in each of the receive memory locations. An example of a diagram of the RX1 and RX2 signals samples (I samples) can be observed in Figure 4.

As soon as the simultaneous process of transmit and receive finishes, another counter starts incrementing the address for the “Read section” of the two RX memories at a frequency referred to as the system clock. The data is delivered to the block that computes cross-correlation.

The sampling clock of 120 MHz is imposed by the USRP platform. The system clock of 30 MHz was chosen in order to satisfy the timing constraints imposed by the FPGA compile engine.

## 5. Baseband Range Compression

In order to determine the delay between the baseband versions of the received signals from both channels, RX1 and RX2, a cross-correlation is computed between the two. Practically, a discrete time domain convolution between a conjugated time-reversed version of the transmitted signal (RX1) and received signal (RX2) is performed. The aforementioned operation is identical to the cross-correlation operation (7).
(7)$$s_{\mathrm{RX}1}\left(n\right)⊗s_{\mathrm{RX}2}\left(n\right)\Leftrightarrow s_{\mathrm{RX}1}^{*}\left(-n\right){*s}_{\mathrm{RX}2}\left(n\right)$$
where ⊗ is the cross-correlation operator and * is the convolution operator.

The definition of the convolution adapted for the involved signals is (8):(8)$$s_{\mathrm{RX}1}^{*}\left(-n\right){*s}_{\mathrm{RX}2}\left(n\right)=\overset{K}{\underset{k=1}{\sum }}s_{\mathrm{RX}1}^{*}\left(k\right)\cdot s_{\mathrm{RX}2}\left(n-k\right)$$
where *K* is the number of multiplications which can be computed simultaneously. 

The baseband range compression operation is performed by computing the cross correlation between a version of the transmitted signal (RX1) and the received signal from the scene (RX2). A simplified block diagram of the cross-correlation implementation is presented in Figure 5. The cross-correlation block works at the system clock frequency (30 MHz).

The most resource-demanding part of the cross-correlator implementation are the multipliers. These multipliers are implemented using “DSP48” slices. Combinational multipliers can also be implemented, but with high hardware resource requirements, which can prove inefficient for the design. A number of 1540 “DSP48” slices are available on the on-board FPGA. Since the multiplications required are in the complexdomain, a number of 3 “DSP48” cells are required for a complex multiplication. Therefore, K was chosen at 448, in order to keep a few spare cells for the rest of the design.

The size of both RX1 and RX2 memories has to be chosen, considering radar system parameters (transmit time duration, maximum detectable range), as well as FPGA resource utilization constraints.

The length of the RX1 signal was chosen equal to the number of simultaneous multiplications (*K* = 448) in order to simplify the implementation. The length of the RX2 memory is chosen at 3136. Since the sampling frequency is 120 Msps, and the address location is incremented every sampling period, the receive window duration is 26.13 μs. Echoes from targets situated as far as 3.9 km can be received in this time window duration.

Since 448 samples of both RX1 and RX2 signals are required simultaneously (at each clock cycle) for the computation of cross-correlation, the memories content must be dumped into structures called buffers. The buffers are practically shift registers composed of flip-flops, with 16-bit width.

The first operation stage consists of dumping the contents of each of the RX1 memories into the corresponding buffer, which is done by decrementing the address counter. Samples shift in the right direction every clock cycle, until the entire content of the memory fills the buffer. Once this dump is finished, the buffer’s implicit shifting is stopped using a disable signal. The imaginary part of the signal (the Q component) is negated before being written to the buffer, in order to perform the complex conjugate of RX1.

Both I and Q components of both signals are subject to mean value subtraction, in order to correct the DC offset introduced by the analog radio front-ends. The mean value iscomputed on the samples of each pulse repetition interval (PRI). At the first PRI, the mean is considered 0. After all the samples have been dumped out of the memory, their mean value is computed. The mean value computation is performed as the arithmetic mean, using a proprietary “LabView FPGA” block. This mean value is memorized and is subtracted from the samples of the next PRI. It was experimentally observed that the mean is very similar from one PRI to another.

After the upper side buffer has been completely filled, the start of the second counter is triggered. This counter increments the addresses of the RX2 memories, therefore filling the downside buffers with samples. After a number 448 of clock periods (equal to *K*, when the first complete overlap of RX1 and RX2 samples takes place), the down counter also issues a “data valid” signal, meaning that the first sample of the computed cross-correlation is ready.

Each complex sample from the first buffer is multiplied with its correspondent from the second buffer, and the results of these multiplications are summed with the adders in the same block, resulting one complex number per each clock cycle. This operation happens at each clock cycle, until the RX2 memory was completely swept through the buffer.

A pseudocode description of these operations can be observed in Algorithm 1:
**Algorithm 1**: Cross-Correlation Computation**Inputs:** S_RX1, S_RX2**Output:** XCORR1for n = 1 to (3136-448) do2  BUFFER2 = flip(S_RX2(n to n + 447))3  for k = 0 to 447 do4    XCORR[n] = XCORR[n] + conjugate(S_RX1[k])*BUFFER2[k]5  end for6end for7return XCORR
where “Buffer2” is the contents of the complex buffer corresponding to RX2 memory at the present clock cycle, and “flip” means horizontal flipping of the vector (the first element becomes the last, the second becomes the last but one, and so on). The operations corresponding to the inner for (lines 3, 4 and 5) are computed in a single system clock period.

As soon as the “data valid” signal goes high, the samples are processed by the next blocks, which compute the magnitude and the phase of the cross-correlation. The magnitude is obtained by summing the squares of the real and imaginary parts, then by computing the square root of the sum. A rectangular to polar converter block is used to compute the phase of the cross-correlation.

Multiplications and summations increase the number of bits of the result. The number of 448 multiplications and 447 summations yield a result with 42-bit width. After the square values computation, the bit width increases at 84 bits, and the summation between the results outputs an 85-bit width. The number of bits is truncated to 64 and the samples of the result are sent to the host PC using two FIFOs. The total FPGA resource utilization can be found in the Appendix A

The cross-correlationimplementation can be performed directly in the time domain, or in the frequency domain by multiplying the computed FFTs of the input signals and then transforming back to the time domain using IFFT. For the time-domain approach, the computational complexity is $O\left(N^{2}\right)$, while for a single FFT block the computational complexity is $O\left(N\log_{2}N\right)$, where N is the length of the input sequence. For the frequency domain approach, there are two FFT blocks, one IFFT block, a complex conjugation block and a multiplication block. The computational complexity of the time domain approach is approximately 17 times higher than that of the frequency domain approach. 

“Labview FPGA” contains intellectual-property blocks for FFT implementation, which allow for a low degree of customizability in order to adjust the latency versus resource usage. Although the usage of these blocks is more computationally efficient, they offer a long response time (latency). Parallelization of these blocks is possible, but not as convenient to implement as the time domain version of the cross-correlation.

Even if the time domain implementation is more computationally complex and more resource-demanding, it was chosen because it can easily be parallelized and obtains faster computation time. Fast computation time is a key design parameter of this system.

Figure 6 shows the timing diagram of the events during a single PRI. The transmit and receive operations start simultaneously. As soon as the receive time finishes, the cross–correlation computation starts. The cross-correlation is computed in 121.63 μs.

The range profile data has the size of 350 kbits, and is transferred from the FPGA to the host computer in 110 μs, according to the specifications of the PCI-Express interface.

The time left until the next PRI is used for the computations performed on the host, which will be described in the next chapter.

The range resolution cell can be determined using the standard formula. Since the chirp bandwidth is 40 MHz, the range resolution cell is 3.75 m. In order to determine the measurement range, a rectangular metal target is considered, with the size of 26 cm × 30 cm, which yields an RCS of 1.55 m^2^ [16]. The bandwidth of the receiver is 160 MHz and the required signal to noise ratio is 10dB, in order to perform displacement measurements with a certain standard deviation, as it will be described in section 7. The radar equation is used in order to determine the measurement range at a value of 92 m.

A summary of the parameters of the implemented system is found in Table 1.

## 6. Host Side Interface

The host computer sets the periodicity of the trigger signal for transmitting the chirp and receives the samples of the range profile (magnitude and phase) from the FPGA. An example of signals obtained after the FPGA processing and displayed on the host computer can be observed in Figure 7.

Additional signal processing is performed on the host PC, using the “LabView” environment. A bin from the profile is selected, corresponding to the monitored target. A target bin is the range correspondence of a round-trip delay time equal to the sampling period *T_s_*. The sampling frequency equals 120 Msps. Therefore, the range bin width is:(9)$$\delta r=\frac{c}{2}T_{S}=1.25\;m$$

Both the real and the imaginary part from the chosen bin are recorded for a specified number of runs. The displacement is computed using Equation (6), with *f*_0_ = 5.755 GHz.

Since the radar is capable of fast signal processing and results display, the complex data in the bin of interest from each of the generated range profiles can be used to “sample” the mechanical motion of the target. By memorizing the variations in the bin of interest at each radar pulse, two relevant plots are obtained: one with the displacement of the target versus pulse number, and another one with the frequency spectrum describing the target’s motion.

A full pack of selectable size is recorded, containing the data in the range bin of interest. The size of the packet equals the number of processed radar pulses. Once the pack is completed, FFT is performed upon the data, in order to determine the vibration frequency spectrum. The horizontal axis can be scaled in frequency of vibration or in velocity. Each FFT result is placed in a waterfall type plot in order to observe the history of the frequency spectrum from the vibrating target.

## 7. Validation and Experimental Results

The proposed system can be used for both displacement and vibration monitoring. Two identical antennas (model MA-WA58-1X, manufactured by “Mars Antennas”, Chicago, IL, USA) mounted on the same pole were used for transmit and receive, oriented in the direction of the target.

### 7.1. Outdoor Displacement Measurement

A corner reflector was mounted as a target and is moved using a linear motion system. The linear motion system features a high precision rotary position encoder, placed on the shaft of the electric motor. This sensor imposes the linear step size precision. Two series of the same scenario were performed, at a radar to target distance of 7.5 m and 30 m, at *SNR* values of 50 dB and 26 dB respectively. A series contains 20 steps, moving the target back and forth between two discrete positions spaced 50 mm apart.

In order to determine the precision for displacement measurements, the user can read the relative change in position of the target from one step to the next one in the graphical interface. The mean values of displacement over 128 pulses are recorded after each step. The differences of the displacement measurement between successive pulses are small and of random nature. Averaging is used in order to mitigate this effect. The phase unwrapping was performed manually. The errors of the measurements can be observed in Figure 8. The standard deviation of the errors recorded for this case is 0.16 mm at most.

An additional measurement set for displacement measurement at the target range of 30 m was performed, but in a different manner: the target was moved with a 5 mm step in the same direction, for a number of 30 steps. The displacement values were unwrapped manually and the results are presented in Figure 9. Currently, there is no automatic phase unwrapping procedure, as described in [17]. The standard deviation of the error is 0.16 mm.

A MATLAB simulation was performed in order to determine the standard deviation of displacement measurement at different signal-to-noise ratio values. A transmitted and a received chirp signal are generated. Noise was added on the received chirp signal, in order to obtain each of the desired *SNR* values, measured on the range profile. The scenario was repeated for a number of 1000 realizations. The displacement is computed using the phase in the range profile bin of interest, with relation (6). The simulated results can be observed in Figure 10.

Additionally, in order to check the results of the simulation, an analytical expression of the standard deviation with respect to *SNR* is plotted. According to [18], the expression is as follows:(10)$$\sigma_{displ}=\frac{\lambda }{4\pi }\cdot \sqrt{\frac{1}{2\cdot SNR}}$$

The analytical values are in good agreement with the simulated ones. As previously described, experimental measurements were performed at two radar-to-target ranges, which yield *SNR* values of 50 dB and 26 dB. The data points are also plotted on the chart of Figure 10, with values close to those of the other traces.

### 7.2. Real Time Target Vibration Spectrum Monitoring

The radar sensor is capable of monitoring the motion of the chosen target and transforming it into vibration spectrum. In order to test this ability, a vibrating target was built.

The vibrating target consists of an audio speaker, which has an FR4 reinforced plate tied to its membrane. The size of the rectangular reflecting plate is 26 cm × 30 cm. The speaker is driven by a general-purpose audio amplifier, whose input is connected to a sinusoidal signal generator. For sensing the amplitude of the induced vibration, a mechanical sensor consisting of a potentiometer was included in the fixture, with the wiper tied to the vibrating plate. A drawing of the entire vibrating target fixture, including the mechanical sensor can be observed in Figure 11.

The vibrating target was placed at a distance of approximately 3 m from the radar. A sinusoidal signal of 12 Hz frequency was applied to the inputs of the speaker. The results of the vibration monitoring can be seen in Figure 12. FFT is performed based on the data captured from a length of 256 pulses. The observed harmonics are presumed to be of mechanical origin.

The vibration frequency measurement accuracy is directly related to the accuracy of the pulse repetition frequency, which is highly dependent on the internal clock accuracy of the USRP platform. The pulse repetition frequency is set by a counter in the FPGA, which increments its value once every sample clock period. The sampling clock is derived from the internal USRP clock, whose accuracy can be increased using a GPS disciplined oscillator. 

The resolution of the FFT spectrum is inversely proportional to the number of radar pulses used in the computation of the vibration spectrum. The number of radar pulses used for the computation of the vibration spectrum is adjustable. The FFT length equals the number of pulses used for the calculation.

### 7.3. Indoor Vibration Measurement

In order to test the precision of the described system at vibration amplitude measurement, the same vibrating target as described in the previous paragraph was placed at close range (3 m) to the radar sensor.

Vibrations of frequencies in the range of 5 to 50 Hz were applied by the speaker to the metal plate, in 5 Hz step increments. The amplitude of the sinusoidal signal at the input of the speaker was kept constant throughout the entire frequency range, but the mechanical oscillations did not yield constant amplitude values due to the mechanical system itself. 

The system measures the real time peak-to-peak value of the displacement recorded for the target bin, in the same manner as described in the first subparagraph of this section.

The comparison between the peak-to-peak vibration amplitudes measured by the potentiometer and the ones measured by the radar can be observed in Figure 13. At low vibration frequencies, there are differences between the peak-to-peak amplitudes recorded by the two sensors of up to 2 mm. The cause of these errors may also be related to the mechanical oscillation modes which appear along the FR4 plate, which yield different vibration amplitudes in the center of the plate (where the measuring sensor is located) than on its edges. Since the target is located in the same range bin, an average value of these amplitudes is actually measured.

The standard deviation of the error between the vibration peak-to-peak amplitude measured using the two methods is 0.6 mm.

### 7.4. Outdoor Vibration Measurement

The same measurement setup was repeated in an outdoor scene, with the distance between the radar and the vibrating target of approximately 10 m. The vibration amplitudes measured by the mechanical sensor were recorded again. The peak-to-peak vibration amplitude values measured by potentiometer and radar can be observed in Figure 14. The standard deviation of the error between the vibration peak to peak amplitude measured using the two methods is 0.6 mm, similar to the one obtained in the indoor measurement set.

The causes of the errors for both indoor and outdoor measurements are probably the spurious changes and the instability of the vibrating target mechanical fixture.

## 8. Conclusions

A radar sensor with the capability of monitoring real time displacement and vibration of remote targets was presented in this paper. The FPGA baseband part and host computer signal processing parts were implemented and tested on indoor/outdoor real-world scenarios.

The main features of this radar are real time range compression and display of the range profiles at fast PRF rates (100 Hz). It is also capable of vibration spectrum monitoring (up to 50 Hz) on a desired target. Since it is a noncontact method of displacement and vibration monitoring for remote targets, it can successfully be used whenever real time display of data is necessary.

The present paper proves that the system is functional in its experimental state. The experiments performed cover specific scenarios. In order to fully characterize the instrument, more experiments are required in order to determine its sensitivity, resolution and measurement range.

The displacement and vibration results offered by the radar were compared to the ones measured with another sensor and showed good agreement. The precision of this radar system is estimated (using the 3σ approximation) at 0.6 mm for displacement measurements and 1.8 mm for vibration amplitude measurements. Other systems meant for displacement measurements have been described in research papers. For example, an interferometric SAR experiment [18] with satellite-borne radar has shown a precision of 0.75 mm (the standard deviation of the displacement errors) for a signal to clutter ratio of approximately 12 dB. Commercial real aperture radar systems such as “FASTGBSAR-R” [19] offer an accuracy of 0.01 mm, as stated by the manufacturer.

In the future, spectral estimation techniques (Capon, MUSIC [20]) could be used instead of FFT on the host processing part.

## Figures and Tables

**Figure 1 sensors-19-00082-f001:**
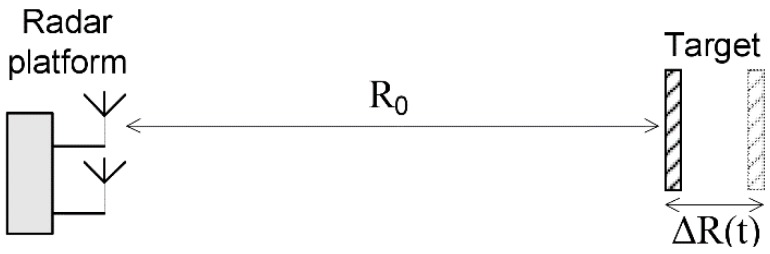
Displacement measurement geometry. Target is placed at a fixed range *R*_0_. The target’s displacement or vibration is described by Δ*R*(*t*). Δ*R*(*t*) is much smaller than *R*_0_.

**Figure 2 sensors-19-00082-f002:**
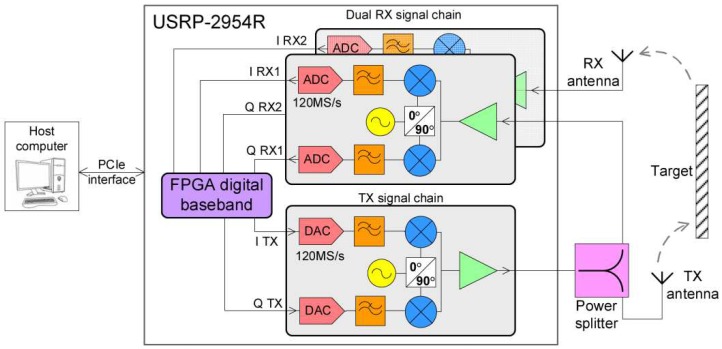
Schematic block of the entire radar system, containing “USRP-2954R”.

**Figure 3 sensors-19-00082-f003:**
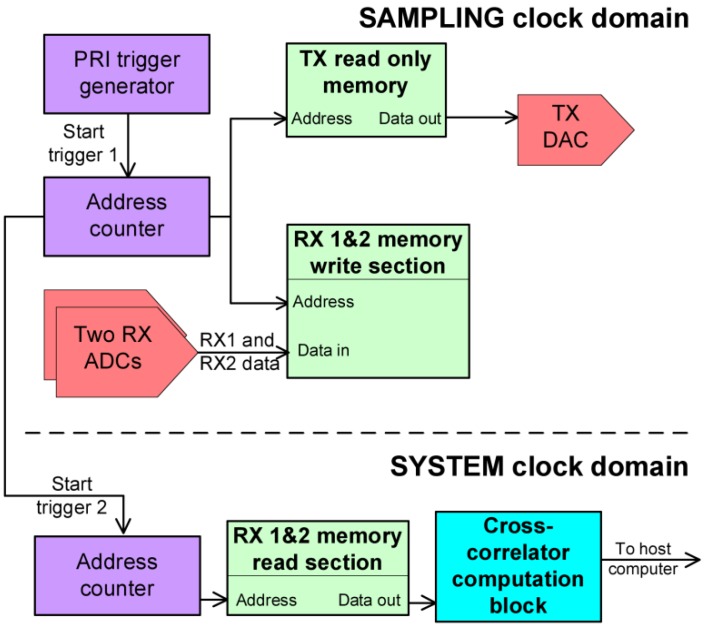
Pulse radar FPGA implementation overview. Two different clock domains exist: system clock and sampling clock. The system clock was chosen based on the implementation timing constraints. The two domains are linked by the “Start trigger 2” signal and by the RX memory contents.

**Figure 4 sensors-19-00082-f004:**
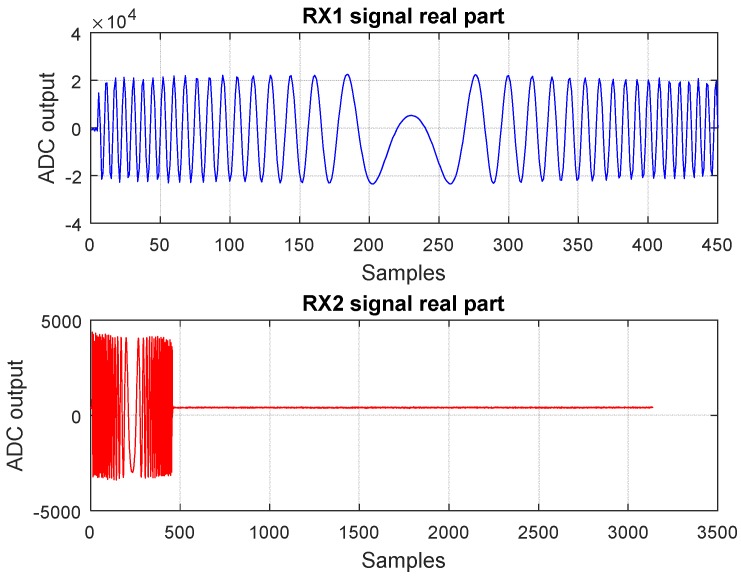
I samples of RX1 and RX2 signals, with their corresponding time domain duration, as received by the platform when a close target is placed in front of the radar.

**Figure 5 sensors-19-00082-f005:**
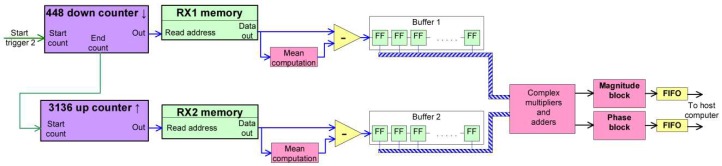
Cross-correlation block diagram, as well as the related blocks that operate in the system clock domain. The contents of the RX1 memory are dumped in the corresponding buffer. Then, the contents of the second memory are dumped in “Buffer 2”. The two buffers are actually shift registers composed of cascaded FFs (flip-flops). The multiplications and partial results summation are performed each clock period. The magnitude and phase of the obtained range profile are sent to the host computer using FIFOs.

**Figure 6 sensors-19-00082-f006:**
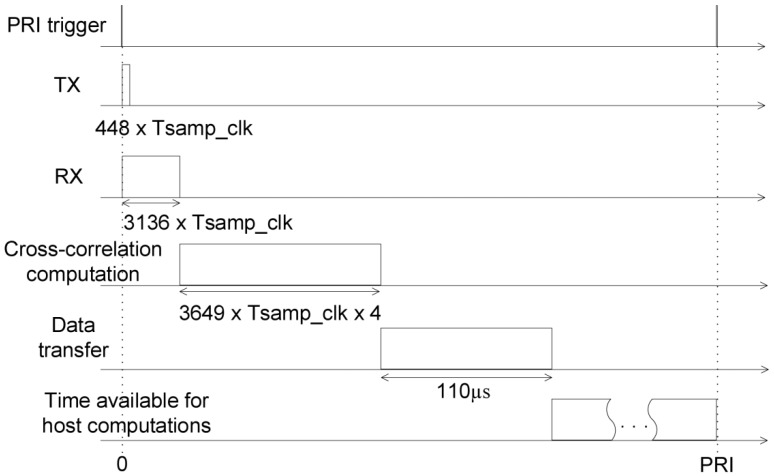
Timing diagram of the operations performed during a PRI. The sampling clock period is denoted Tsamp_clk and the system clock period is denoted Tsys_clk. The system clock period is four times greater than the sampling clock period. The time available for data transfer from FPGA to host and additional host processing is the time left after the end of cross-correlation computation until the next PRI.

**Figure 7 sensors-19-00082-f007:**
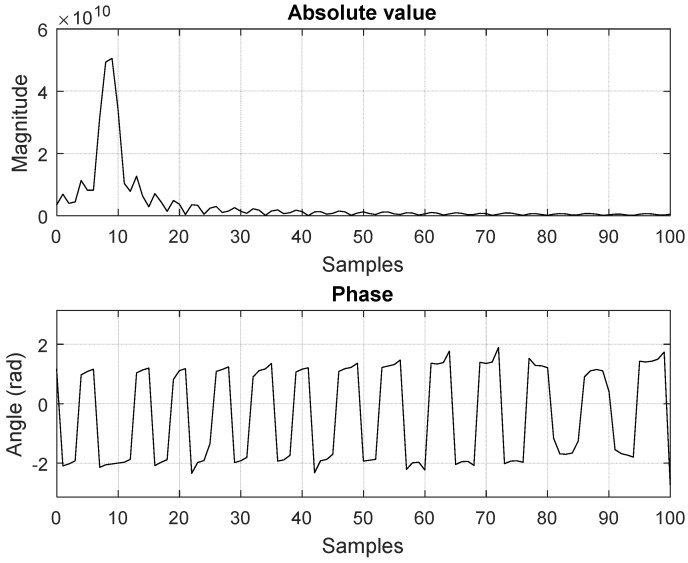
Resulted signals from FPGA range compression, based on the RX1 and RX2 signals plotted in Figure 4, as displayed on the host interface. The peak in the first plot indicates the delay between the two signals.

**Figure 8 sensors-19-00082-f008:**
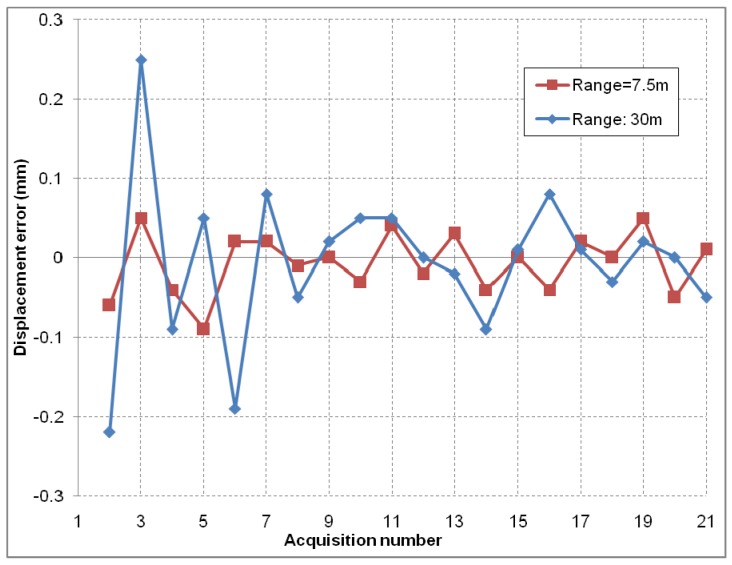
Displacement measurement errors at 7.5 m and at 30 m range. Note the slightly larger error for the 30 m range.

**Figure 9 sensors-19-00082-f009:**
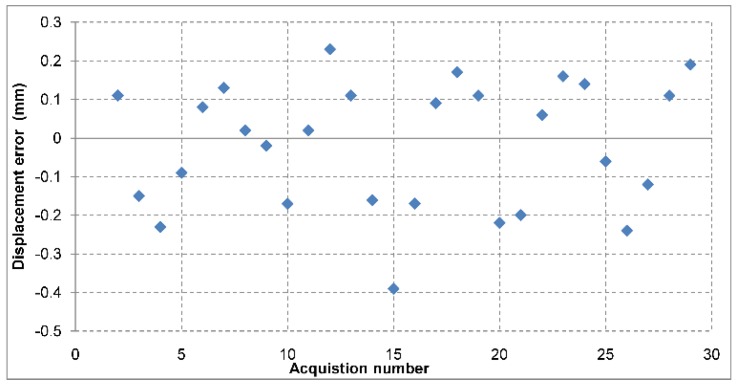
Displacement measurement errors at 30 m range for 30 consecutive steps in the same direction, with a step size of 5 mm.

**Figure 10 sensors-19-00082-f010:**
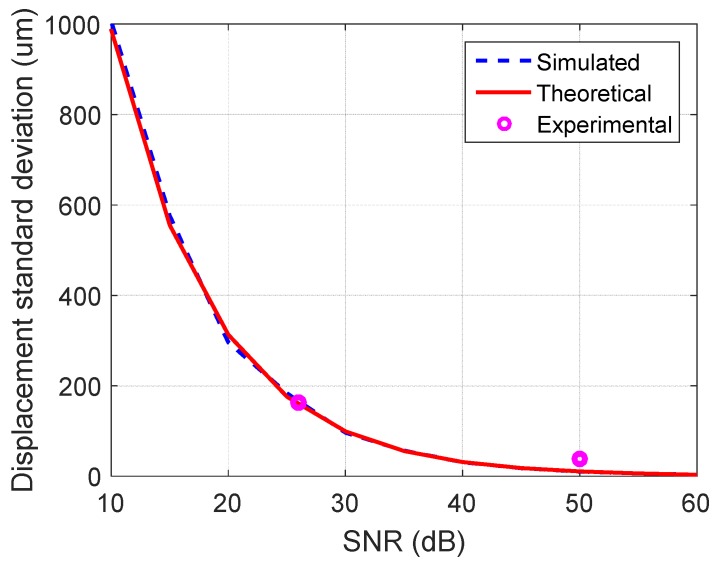
Standard deviation of the displacement measurement versus signal-to-noise ratio, in the simulated, theoretical and experimental cases.

**Figure 11 sensors-19-00082-f011:**
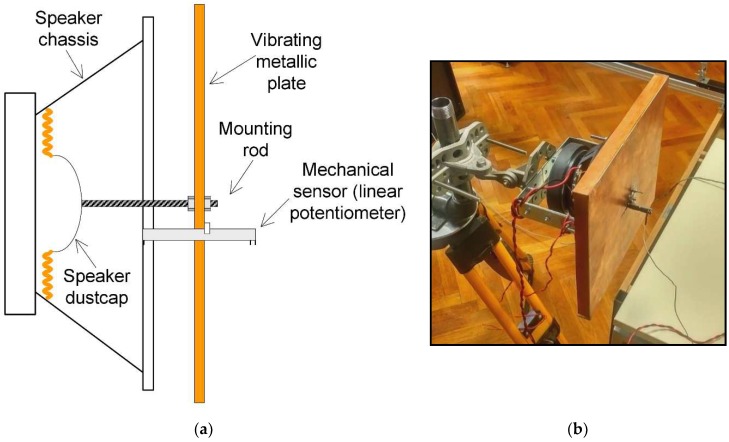
(**a**) Vibrating target mechanical fixture drawing. An audio speaker is used as the vibrating element, with a metallic plate mounted on the dust cap. A linear potentiometer is used to sense the vibration amplitude. (**b**) Photography of the assembled mechanical fixture mounted on a tripod.

**Figure 12 sensors-19-00082-f012:**
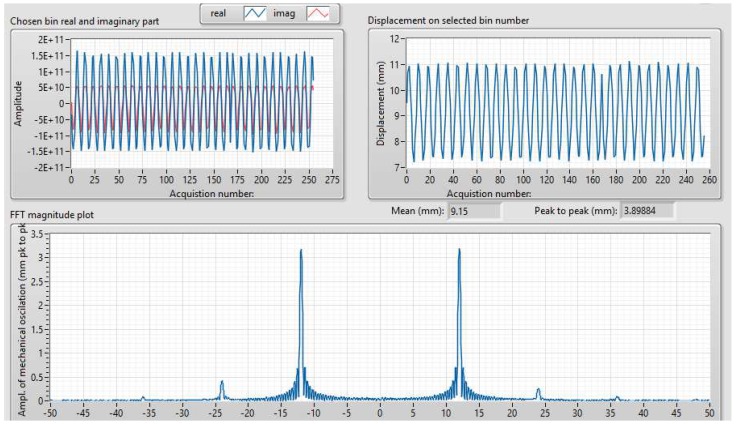
Vibration monitoring on the vibrating target, set at a 12 Hz frequency. Data from 256 radar pulses was used for the computation of this spectrum. The fundamental component at 12 Hz is the most powerful, followed by weak harmonics at 24 Hz and 36 Hz.

**Figure 13 sensors-19-00082-f013:**
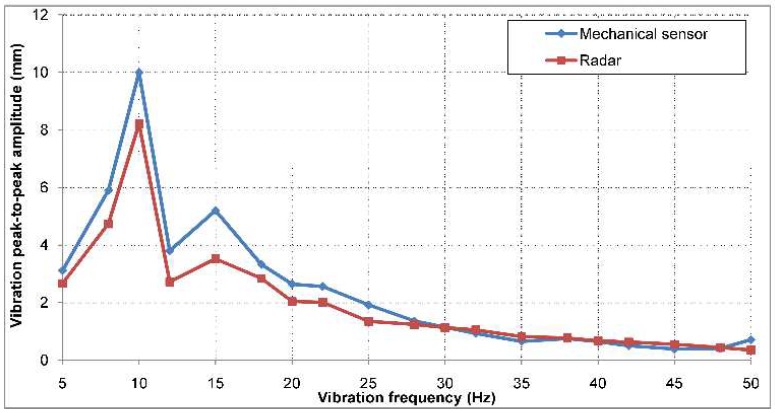
Indoor vibration amplitude measurement results, for a frequency range from 5 Hz to 50 Hz (half the PRF of the radar, in order to prevent aliasing). The indoor target was placed at a range of 3 m.

**Figure 14 sensors-19-00082-f014:**
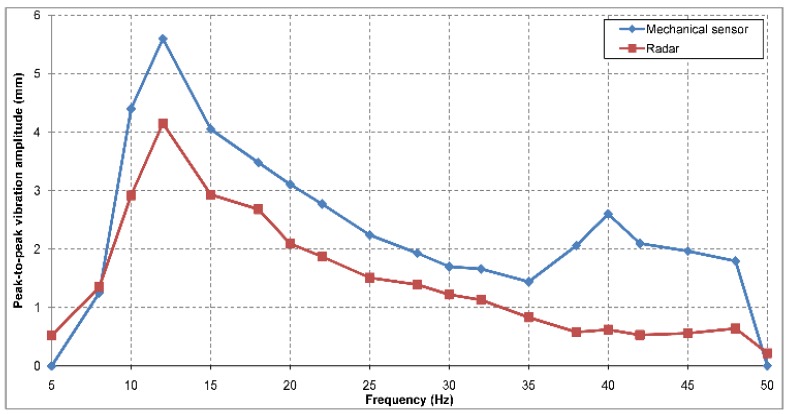
Outdoor vibration amplitude measurement results, for a frequency range from 5 Hz to 50 Hz. The radar to target range was approximately 10 m.

**Table 1 sensors-19-00082-t001:** Parameters of the implemented radar system.

Radar System	Parameters
Carrier frequency	5.755 GHz
Transmitted power	30 dBm
Chirp bandwidth	40 MHz
Range resolution cell	3.75 m
Maximum detectable target range	92 m
Transmit pulse duration	3.73 μs
Receive window duration	26.13 μs
Cross-correlation computation time	121.63 μs
Pulse Repetition Interval	10 ms

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
