# Peer review of "Pulse Radar with Field-Programmable Gate Array Range Compression for Real Time Displacement and Vibration Monitoring"

_sensors, 2018, doi:10.3390/s19010082_

Reviewer 1 Report

The paper's motivation and contribution are clear. The work has a relevant contribution to the state. The exposition is generally clear and at an appropriate level of detail. 

The main missing aspect is a systematic characterization of the instrument (resolution, measurement range, precision as a function of distance to target). This point is actually acknowledged in the conclusions, but should nevertheless be addressed.

Clarifications should also be included regarding some figures:

Fig. 13: what could be the cause of the behavior at 50 Hz (or is it just a spurious change)?

Fig. 14: behavior of the mechanical sensor between 35 Hz and 50 Hz?

Minor points:

State clearly what is understood by "bin" and how they are determined/defined.

Definitions of the abbreviations FMCW and PRF are missing.

In several places, "bit depth" is mentioned wheras "bit width" would be correct.

(page 8) "pack" should perhaps be "packet"

(fig. 8) "Acquistion" --> "Acquisition"

(fig. 11) Hard to read on paper (ok in the pdf)

algorithm 1: indentation level of  the "for" loop

Reviewer 2 Report

Dear authors,

I reviewed the paper entitled “Pulse radar with FPGA range compression for real-time displacement and vibration monitoring” by Mihai Tudose et al. The paper describes a new radar platform for real-time   monitoring of displacement and vibration of remote targets. The authors   describe different methodologies for this kind of monitoring systems   analyzing their different shortcomings and the advantages of the proposed   platform. The description of the system is correct and explained in a   comprehensible manner.

Although the instrument is in an experimental stage, it   proves to be efficient and functional in the proposed experiments, being   validated with other sensors. However, different aspects have to be assessed   such those described in the conclusions before using the platform for   monitoring in a more operational way. The authors are aware of this analysis   that could be proposed for future work where the radar sensor is applied to   different scenarios. However, these aspects could be briefly discussed to   assess the potential of this platform for real-time monitoring.

The paper is well written with a correct English language.   The structure of the paper is also correct and the style makes a fluent   reading.

Minor points:

In the abstract also indicate the 3-sigma values of the   precision: “The precision (3s) of   this system…”

Unify the use of “real-time” vs. “real time” along the   manuscript including the title.

Lines 82 and 92, write “Where” with lower case and at the   beginning of the sentences as in lines 86, 164 and some others.

Line 128. Write “Schematic block of” instead of “Block   schematic of”.

Enlarge figure 12 as it is difficult to read.

Reviewer 3 Report

Overall, this is a somewhat interesting description of a prototype for radar-based vibration monitoring, based on FPGA architecture. However, before publication, it is my opinion that the authors should address the following major comments:

1)  Please define all the acronyms for completeness, e.g. USRP and FPGA.

2)   On the statement “By means of Doppler effect or by means of interferometry, radars can also be used in non-contact vibration monitoring” -> This applies also to sonars, e.g.

[R1]  "Low observable target motion analysis using amplitude information." IEEE Transactions on Aerospace and Electronic Systems 32.4 (1996): 1367-1384.

[R2]  "Tracking the tracker from its passive sonar ML-PDA estimates." IEEE Transactions on Aerospace and Electronic Systems 50.1 (2014): 573-590.

Please mention this application also for completeness.

3) Please add a notation paragraph at the end of Sec. I for the sake of completeness.

4) The related works part should be improved in my opinion.

5) In my opinion, a discussion on the computational complexity of the proposed system would be extremely desirable.

6) I would encourage the authors at least discussing the applicability of the proposed system also to detection purposes. Also, it would be useful discussing the applicability of the proposed system to estimation of multiple vibration components following “imaging” approaches, e.g.

[R3] "Time reversal imaging of obscured targets from multistatic data." IEEE Transactions on Antennas and Propagation 53.5 (2005): 1600-1610.

[R4] "Performance analysis of time-reversal MUSIC." IEEE Transactions on Signal Processing 63.10 (2015): 2650-2662.

7)      Some figures are hard to read and should be improved in terms of clarity and readability.

8)      Please enrich conclusions with further avenues of research.

Reviewer 4 Report

REVIEW

Article titled “Pulse radar with FPGA range compression for real time displacement and vibration monitoring”

Sensors no. 403228

List of Authors

Mihai Tudose, Andrei Anghel, Remus Cacoveanu, Mihai Datcu

Authors in this paper proposed the basic functionality of a radar platform for real-time monitoring of displacement and vibration. Introduction contains a very extensive revision of the world bibliography concerning different methods of vibration monitoring of remote targets, signals processing for real-time vibration monitoring of targets and coherent integration and application for vibration monitoring. Presented in Sections 2 (Principle of pulse radar displacement monitoring ) and 3, 4 5, i.e. System overview, Pulse radar baseband implementation on the USRP platform and Baseband range compression are well-defined and described. Used in this research paper the mathematical formulas and equations seem to be correct.

My direct remarks concerning this article.

1. Some experimental results are illustrated in Figure 4. In Figure 4 there is no vertical axis designation - should be improved.  The same situation occurs in Figure 7 – should be improved.

2. Figure no. 12 is too small. It's hard to see anything - should be enlarged.

3.In my opinion, considering the different technologies applicable in radars, parameters of emitted radar signals, e.g. different types of PRF modulations (sliding, stagger, dwell and switch, jitter) and next possibilities their identification, SAR signal processing - the following papers are also supposed to be listed in the References:

 Fast-decision identification algorithm of emission source pattern in database. Bulletin of the Polish Academy of Sciences, Vol. 63, No. 2

Optimizing the Minimum Cost Flow Algorithm for the Phase Unwrapping Process in SAR Radar. Bulletin of the Polish Academy of Sciences, Vol. 62, No. 3

An application of iterated function system attractor for specific radar source identification. 17th International Conference on Microwaves, Radar and Wireless Communications MIKON-2008, vol. 1, pp. 256-259.

4.   In order to validate the proposed method the simulations are conducted using “USRP-2954R” platform, the host computer and “LabView” software.

There is also no comment from the authors on what the computational burden of the proposed method is and if their solution is used in equipment working in real conditions. Their method may be in the analytical stage in a simulated computational environment –  it should be elaborated and explained without any doubt.

5.   The authors are not willing to write a presentation in what way the proposed method is better than other methods which can be found in the References. In conclusions there is a lack of precise analysis of the received results in comparison with other SEI methods. The article is supposed to have such a comparative analysis of the received results.

Conclusion:

The work should be reviewed after completing it with all necessary answers to questions above as well as required comments and information

Author Response

Please check the attached document.

Round  2

Reviewer 1 Report

The revised version addresses my previous comments.

Minor point:

- (page 8): "at it will be"  -> "as it will be"

Author Response

Thank you for the rectification. It has been addressed.

Reviewer 3 Report

Overall, this is a somewhat interesting description of a prototype for radar-based vibration monitoring, based on FPGA architecture. Additionally, the authors have satisfactorily addressed all my previous comments and modified the manuscript accordingly. I have only the following remaining minor comment that I would like the authors to address when preparing the proof files:

Please add a notation paragraph at the end of Sec. I, i.e. collecting all the mathematical symbols employed throughout the paper, for the sake of completeness.

Author Response

Thank you for your suggestion. A paragraph containing all the mathematical notations from the paper has been added at the end of section I.

Reviewer 4 Report

REVIEW_2

Article titled “Pulse radar with FPGA range compression for real time displacement and vibration monitoring”

Sensors no. 403228

List of Authors

Mihai Tudose, Andrei Anghel, Remus Cacoveanu, Mihai Datcu

The article Sensors no. 403228 entitled “Pulse radar with FPGA range compression for real time displacement and vibration monitoring” has been carefully modified and well revised.

The work is supposed to be finally accepted for publication in Sensors.

Author Response

Thank you for the review process and suggestions.